# Agro-Techniques for Lodging Stress Management in Maize-Soybean Intercropping System—A Review

**DOI:** 10.3390/plants9111592

**Published:** 2020-11-17

**Authors:** Ali Raza, Muhammad Ahsan Asghar, Bushra Ahmad, Cheng Bin, M. Iftikhar Hussain, Wang Li, Tauseef Iqbal, Muhammad Yaseen, Iram Shafiq, Zhang Yi, Irshan Ahmad, Wenyu Yang, Liu Weiguo

**Affiliations:** 1Key Laboratory of Crop Ecophysiology and Farming System in Southwest China, Ministry of Agriculture, Sichuan Agricultural University, Chengdu 611130, China; 2018601001@stu.sicau.edu.cn (A.R.); 2018201013@stu.sicau.edu.cn (C.B.); S20176512@stu.sicau.edu.cn (W.L.); alisicau.edu@gmail.com (T.I.); sicau.edu.433@gmail.com (I.S.); 2019201010@stu.sicau.edu.cn (Z.Y.); 2019605002@stu.sicau.edu.cn (I.A.); mssiyangwy@sicau.edu.cn (W.Y.); 2Institute of Ecological Agriculture, Sichuan Agricultural University, Chengdu 611130, China; 3CAS Key Laboratory of Mountain Ecological Restoration and Bioresource Utilization & Ecological Restoration Biodiversity Conservation Key Laboratory of Sichuan Province, Chengdu Institute of Biology, Chinese Academy of Sciences, University of Chinese Academy of Sciences, Wuhou 610000, China; ahsanasghar2017@mails.ucas.ac.cn; 4Department of Plant Breeding and Genetics, University of Agriculture, Faisalabad 38000, Punjab, Pakistan; L201702011@stu.sicau.edu.cn; 5Department of Plant Biology & Soil Science, Universidad de Vigo, 36310 Vigo, Spain; mih786@gmail.com; 6State Key Laboratory of Crop Gene Exploration and Utilization in Southwest China, Institute of Rice Research, Sichuan Agricultural University, Wenjiang, Chengdu 625014, China; 2018511004@stu.sicau.edu.cn

**Keywords:** intercropping, lodging tolerance, agronomical management, lignin metabolism, resistance genes

## Abstract

Lodging is one of the most chronic restraints of the maize-soybean intercropping system, which causes a serious threat to agriculture development and sustainability. In the maize-soybean intercropping system, shade is a major causative agent that is triggered by the higher stem length of a maize plant. Many morphological and anatomical characteristics are involved in the lodging phenomenon, along with the chemical configuration of the stem. Due to maize shading, soybean stem evolves the shade avoidance response and resulting in the stem elongation that leads to severe lodging stress. However, the major agro-techniques that are required to explore the lodging stress in the maize-soybean intercropping system for sustainable agriculture have not been precisely elucidated yet. Therefore, the present review is tempted to compare the conceptual insights with preceding published researches and proposed the important techniques which could be applied to overcome the devastating effects of lodging. We further explored that, lodging stress management is dependent on multiple approaches such as agronomical, chemical and genetics which could be helpful to reduce the lodging threats in the maize-soybean intercropping system. Nonetheless, many queries needed to explicate the complex phenomenon of lodging. Henceforth, the agronomists, physiologists, molecular actors and breeders require further exploration to fix this challenging problem.

## 1. Introduction

Climatic change and population explosions are the major threats to food security in the future [1,2]. It has been projected that, by 2050, the present world population will be enhanced by up to 30%, which will make the world population nine billion or more people [3]. This problem can only be solved through multiple cropping systems to fulfill the food demand and supply requirements which leads to sustainable agriculture [4]. The maize-soybean intercropping is one of the most important systems that plays a key role in the sustainability of food production systems [5]. The maize-soybean intercropping system has great importance among the legume-cereal intercropping systems because of its maximum yield and efficient use of resources [6,7]. Initially, it was developed in the South-Western region of China and now it is progressing throughout the globe. It is estimated that from the total cropped area of 182.3 million hectares, about 83% area is used for intercropping in Africa [8]. In China, half of the total grain yield is gained through multiple cropping systems [9]. The maize-soybean intercropping system has been adopted in different parts of China due to its maximum production and land use efficiency [7]. The biggest challenge of this century has to be met by China to boost the production of cereals by approximately 600 Mt by 2030 to achieve food security [7]. In another study, it is reported that annually 2.8–3.4 × 10^7^ ha area was grown under the intercropping system in China [10]. Over 667 thousand hectares of soybean are being intercropped with maize in the south-west of China [11,12].

Although this system has many advantages, in this intercropping system the spatial light pattern affects the growth of soybean due to the shading of maize plants during the co-growth period [13]. In order to capture more light, the soybean plants increase their heights and this phenomenon is called shade avoidance. Shade avoidance causes several morpho-physiological changes such as low photosynthetic activities, increased intermodal length, decreased stem diameter and higher rate of lodging [14]. Previous studies revealed that continuous and periodic prevailing of shade had decreased the total grain production, approximately 25–45% [15,16]. However, the climatic factors such as continuous storms and heavy rainfall comprise about 8% and 19% lodging to crops, respectively [17,18]. Many studies focused on the monocropping conditions of the soybean, fertilizer regulation and lodging resistant cultivars [19,20]. In addition, higher N application rates could enhance the lodging threat due to excessive canopy growth and stem elongation [21]. It is also noticed that excessive canopy growth decreases the light interception which in turn elongates the stem length [22,23]. Chen et al. [24] briefed that higher application of basal N decreased the lignin content in internodes. Moreover, it has been seen that high planting densities also reduced the lignin content in stem which leads to weaker stem and hence causes the plant lodging [25].

It has been described that the lodging of stem significantly impeded the photosynthetic activities of the plant at the grain filling stage [26]. Some authors reported that Silicon (Si) had enhanced the cell wall thickness of rice stems, shortened the internode length, changed the canopy structure of plant, increased the photosynthesis and hence prompted the lodging resistance [27]. It has been observed that Si improved the stem strength of rice at the reproductive stage [28] and also increased the content of soluble sugars in maize [29]. Another speculation had revealed that the lodging angle at 25–90° from the right angle could decrease the grain production by approximately 20–61% during the grain filling stage in wheat [30]. Shading is an inevitable factor in intercropping systems; many researches have been conducted to mitigate the adverse effects through breeding of shade tolerant cultivars of soybean [31]. However, developing the shade resistance potential of existing cultivars and clarifying the resistance of different genotypes to shade, is one of the most economical and proficient ways to resolve this problem. It has been reported that compounds of Ti can promote the growth of crops, biomass, enzyme activity, chlorophyll content, iron (Fe) uptake and also compensated the nitrogen deficiency [32]. In addition, it was observed that Ti nanoparticles transformed the expression of miRNAs 16 and modified the root architecture [33,34]. The relationship between the heritability of lodging resistance and the height of plant or seed yield has been elucidated previously [35]. In studies of multiple populations, a modest to high lodging resistance heritability has been examined [36]. Although intensive QTLs (Quantitative Trait Loci) techniques have directed to enhance the lodging resistance in soybean [37]. It is extensively reported that shade avoidance had increased the lodging rate in the maize-soybean intercropping system. However, lodging depends on the stem breaking strength, anchorage of roots (root lodging) and structural carbohydrates accumulation in the stem [38,39].

To date, many studies have been conducted about the relationship between lignin biosynthesis and stem strength in intercropping but the approaches that explore how to manipulate the lignin metabolic pathway that enhance the lodging resistance of intercropped soybean have not been elucidated yet. This review endorsed the recent approaches and future prospects to enhance the stem mechanical strength and lodging resistance of soybean under the maize-soybean intercropping system. Therefore, the following modern agronomical, genetics, chemical and genomic approaches are described in this review; (i) to mitigate the shade effect of tall stature plants on the small plants, (ii) how to create a better light environment for intercropped soybean? (iii) to reduce the stem elongation and increased the soybean stem diameter (iv) to enhance the structural carbohydrates contents, which ultimately increased the lodging resistance and yield of soybeans under the maize-soybean intercropping system. 

## 2. Lodging; A Serious Threat to Intercropping System

The maize-soybean intercropping system is widely adopted in the south-west of China. Over 667,000 hectares area is under soybean maize intercropping system [11]. However, there are still some drawbacks in this system likewise; taller plants (maize) shaded the short stature plants (soybean) during the middle and later growth stages of soybean [12]. Shading of maize caused the stem elongation, slender stem and lower amount of lignin content in the stem which resulting in the soybean lodging [19,40]. It is previously documented that, lodging is one of the main factors that result in the reduction of soybean yield up to 50% [41].

### 2.1. Stem Development and Lignin Metabolism under Intercropping

In intercropping system, the plant height, petiole length and internodal length of soybean plant increased and stem diameter decreased due to the more allocation of carbon to the stem and petiole elongation instead of leaves and roots development [42,43]. This mechanism helps the plants to escape and found more light which results in increasing plant height and low stem diameter [44,45]. Furthermore, it was documented that shade conditions or low intensities of light under intercropping, change the sheath of vascular bundles and stem mechanical layers of tissues which play an important role in the stem anatomical structure to increase plant lodging resistance [46].

Another study had shown that shading of maize had increased the length of soybean stems by up to 45.75% that resulted in lodging and ultimately reduced the final yield up to 20–40% [47]. It has been described that the lodging of stem significantly hindered the photosynthetic activities of the plant at the grain filling stage [26]. It is also reported that shade reduces the area of xylem, pith and ratio of xylem tissues [48]. Moreover, it also decreases the stem cross-sectional area, number of vascular bundles and thickness of stem skin [49]. Lignin consists of complex aromatic polymers and constitutes the cell wall of vascular plants [50]. Previous studies observed that cell wall mechanical strength was increased through crosslinking of hemicellulose and cellulose polymers [51]. Under shade conditions the enzymatic activities that is, cinnamyl alcohol dehydrogenase (CAD), phenylalanine ammonialyase (PAL), peroxidase (POD) and 4-coumarate: CoA ligase (4-Cl) were limited which resulted in lower lignin production during lignin biosynthesis [52,53].

A significant and positive correlation of plant height with yield and yield associated parameters has been reported [54]. Previous studies also found that higher lodging percentage with weaker stems caused reduction in the transportation of carbohydrates and dry matter which resulted in the enhancement of the risk of pest and pathogen attacks that resulted in lower yield [55]. However, it was documented that non-structural carbohydrates are moved to seeds during the seed filling stage for seed formation, the mechanical stability of plants depends on the structural carbohydrates (cellulose and lignin) in the lower part of the stems [56]. Stem development relies on the primary constituents of cell wall (lignin and cellulose) and they significantly correlated with stem mechanical strength and lodging resistance [44]. It has been described that the lodging of stem significantly hindered the photosynthetic activities of the plants at the grain filling stage [26]. The consequence of stem lodging caused a reduction in grain production, increased the harvest cost and lower the quality of grains [30]. Another study revealed that 80% of the grain production reduced due to the impact of lodging naturally and artificially on the total crop yield [57].

Stem development and lodging resistance also depend on the various environmental and field management factors as depicted in Table 1. Moreover, the stem mechanical strength is influenced by non- structural carbohydrates and structural carbohydrates mainly cellulose and lignin [58,59]. The accumulation of lignin and cellulose in the stem had significantly enhanced the stem mechanical strength resulting in the increased lodging resistance which is previously reported in rice, wheat and buckwheat stems [25,59]. The lignin metabolic pathway depends on the chemical activities of lignin related enzymes PAL, 4-Cl, CAD and POD [60,61]. The lignin biosynthesis depends on the genotype and environmental changes, likewise, a significant correlation has been found between the lignin biosynthesis and lignin related enzyme activities (PAL, CAD, POD) in the stems of different lodging resistance wheat and buckwheat cultivars [62,63]. Shade impacts the lignin biosynthesis by affecting the activities of lignin related enzymes in the metabolic pathway of lignin [64,65,66]. However, further studies revealed that shade resistant cultivars have maximum activities of lignin related enzyme (POD, 4-Cl, CAD and PAL) as compared to shade sensitive cultivars [59,67]. Furthermore, it was found that shade had slowed down the activities of lignin related enzyme that caused a lower lignin accumulation in stem of soybean and ultimately leads to weaker stem and lodging [59].

As reported in earlier studies that the accumulation of cellulose and lignin in lodging resistant varieties of rice were higher than that in lodging susceptible varieties of rice varieties [62,68]. It has been found that shading caused the stem elongation and weaker stem which decreased the plant mechanical strength [69]. Shading had also amplified the rate of lodging and conversely increment in the intensity of light had improved the stem strength and lower the rate of lodging in maize [70]. Lodging tolerance can be increased by reducing the plant height and lower the center of gravity point and decreased the aboveground weight of the plant on the basal stem [71,72,73]. Although all the above posted findings had given some clues to understand the lodging mechanism in the intercropping system, still there is a big gap of information that needs to be explored yet to get better insights on the lodging phenomenon and its management under intercropping systems.

### 2.2. Role of Carbohydrates and Lignin Biosynthesis

Lignin and cellulose, the structural carbohydrates, are the main constituents of the cell wall and their components play a vital role in plant vigor, stem strength and the lodging resistance of plants [80]. High lignin concentrations in vascular bundles can increase the strength of the cell wall and boost up the physical resilience of plant stem. The overall lignin content in the basal second internode was significantly correlated with stem strength and elasticity [25,71]. During the development of secondary cell walls, lignin is deposited in the carbohydrate matrix of the cell wall, which makes the whole plant body rigid and allows the plant to develop upwards [81,82]. In addition, by increasing the physical strength of the stem the threat of lodging could be minimized. Lignin was considered to be a macromolecule that played a supporting and fundamental role in enhancing the stem mechanical ability, increasing stem strength, maintaining stem stability and ultimately decreasing the lodging rate by preserving the stem verticality [83,84,85]. Previous research data revealed that the lignin deposition in the stem was positively correlated with stem breaking strength per section area which indicated that a high concentration of lignin in stem had enhanced the stem physical strength. Furthermore, it was revealed that the activities of the enzymes, that is, PAL, POD, CAD and 4CL played a key role in lignin metabolic pathway [86].

Lignin biosynthesis mainly depends upon the multiple enzymatic activities. Among the lot, the PAL enzyme is one of the most important enzymes that plays a role as rate limiting enzyme and catalyst which convert the L-phenylalanine dehydrogenase into trans-cinnamic acid in the shikimic acid pathway [76,87]. CAD is another important enzyme that takes part in the final reaction of the reduction of lignin biosynthesis [61]. It is revealed that the lignin is biosynthesized through lignin monomers polymerization through POD activity and oxidation reaction of monolignols carried by peroxidase [88,89]. The previously conducted experiment had described the significant positive correlation between the activities of lignin related enzymes (PAL, POD, CAD and 4CL) and lignin content in stem of soybean [55]. Furthermore, some studies had also shown that lodging resistance of plants and their varying degree of lodging tolerant is significantly depended on the level of genes transcript abundance and their contribution rate to lignin related enzymatic activities that enhanced the stem mechanical strength [90]. However, recent research had found that a strong correlation between the expression levels of cinnamoyl-CoA reductase2 *(CCR2)*, ferulate 5- hydroxylase *(F5H2)*, caffeic acid O-methyltransferase2 *(COMT2)*, p-coumarate 3-hydroxylase *(C3H1)* and 4-coumarate: CoA ligase1 *(4CL1)* and lignin contents that enhanced the lodging resistance in wheat [91]. In addition, it is also described that auxin deposition in *Arabidopsis* was prompted by hyper-gravity which led to particular lignin metabolic genes expression at a high level and successively led to the process of lignification in stem inflorescence [92]. Moreover, it has been reported previously that auxin and cytokinins up-regulated the genes expression to lignin related enzymes and secondary cell wall growth and development/lignification [93]. However, in maize–soybean intercropping, more research from biologists and physiologists is still required to understand the shading phenomenon; how shade stress of tall stature crops affects the biochemical and physiological activities of short stature crops that related to lignin biosynthesis and lodging resistance? On the other hand, some biotic and abiotic factors including cellular signaling such as hormones activities that play an essential role in the up-regulation of lignin related enzymes and lignin biosynthesis (i.e., lignin and auxin relationship) under the maize-soybean intercropping system have not been elucidated yet (Figure 1). 

### 2.3. Plant Hormonal Activities and Shade Avoidance under Intercropping SYSTEM

Plant hormones are the major elements that take part in the regulation of many plant characters which have a vital role in lodging resistance [94,95,96]. It has been reported that shade avoidance is a complicated phenomenon that regulates the metabolic and transcriptional factors and ultimately prompted the elongation of stem and apical dominance which in turn supports the young tissues escaping from shade [44,97,98]. Some plants evolve and adapt the shade tolerance strategies to counter the shading effect and enable them to survive under low light conditions [42]. The shade avoidance had enhanced the lodging rate in maize-soybean intercropping system, however, lodging depends on the stem breaking strength, anchorage of roots (root lodging) and structural carbohydrates accumulation in the stem [38,39]. WU Yu-shan et al. investigated the relationship between shade avoidance responses and yield analysis of various soybean cultivars under the relay intercropping system [78]. They concluded that under shady conditions the stem length and specific leaf area were increased by 0.78 and 0.65% as compared to full light conditions. On the other hand, the diameter of stem, leaf area, total biomass, number of nodes and branches were reduced than that of normal light conditions [99]. However, the scientists have no comprehensive knowledge yet about the relationship mechanism between the hormones and shade avoidance response under maize-soybean intercropping. In addition, shade avoidance response is also a crucial factor that affects the normal growth of crops in dense canopies [100,101]. In a recent study, Cui et al. [102] investigated the effect of Gibberellin (GA) application on the maize hormonal and antioxidant activities and grain filling under high planting densities. They illustrated the significant correlation between GA application and the level of endogenous hormones and antioxidant activities of maize under high planting densities. They further found that the exogenous application of GA had enhanced the antioxidant contents (SOD, CAT, MDA and POD) and hormones level (IAA, ABA, ZR and GA3) and finally grain yield of high density maize. Furthermore, it is also investigated that the exogenous application of indole-3-acetic acid (IAA) repressed the bud growth of tillers while the zeatin (Z) hormone significantly promoted the growth of buds under low nitrogen environments [94]. The results of their study recommended that Z application had a strong influence on the tillers and tillers buds growth regulation, therefore Z application supports the strong soil anchorage of plants and ultimately creates a more lodging tolerance environment for plants. Nevertheless, the relationship between phytohormones (auxin) and lignin in the maize-soybean intercropping system is still uncertain yet (Figure 2).

## 3. Agro-Techniques for the Management of Lodging Stress

### 3.1. Genetic Manipulation for Increasing Lodging Resistance

The poor resistance to lodging could reduce the soybean yield potential. Previously, independent studies have indicated a significant number of observations of quantitative trait loci (QTL) for lodging resistance [103]. A recent investigation on the integration of QTLs in the lodging resistance of soybean indicated the four QTLs which resulting in the two considerable QTL integrations on chromosomes 6 and 19. Their finding could be useful to increase the lodging resistance in soybean. Their results find a strong and pleiotropic relationship between the lodging resistance and QTL integration on chromosome 6 [104]. Several genes and their QTLs revealed the resistance to lodging and its related traits have been reported in rice (4CL gene family), wheat and barley [105,106,107]. Along with conventional breeding, we have to focus on the identification of lodging-resistance genes especially for cereal crops [108]. Now, identification and transformation of desired genes are much easier because of recent advances in breeding, genomics and biotechnology, which eventually help to increase crop productivity [109]. So, the transformation of lodging susceptible genes to lodging resistance genotypes has the potential to increase the grain yield of cereal crops under lodging prone zones [110]. The population based studies have been reported for QTL mapping in cereals and exhibited lodging resistance especially in wheat [111]. The QTLs study of phenotypic traits which are directly associated with lodging resistance in cereals had already been reported like plant stalk strength, pith diameter, culm diameter and culm wall thickness in wheat [112].

These QTLs have significant effects on lodging but further validation is needed by fine mapping or other advanced techniques. Currently, some of them have already been validated, for instance, in winter wheat a dominant Rht5 gene related to dwarfism in plant height, has been marked on chromosome 3BS linked with molecular marker Xbarc102 [113]. The recent studies revealed that this region on chromosome 6A showed higher phenotypic variations for plant height and could be used further for QTL mapping studies [114,115]. *Rht3* is another dwarfing gene in wheat which shortened plant height up to 59% but has not been used in commercial varieties [116]. *Rht5*, a plant hormone (GA)-responsive gene for dwarfism, which significantly shortened the plant height up to 55% without having negative effects over coleoptile length and seedling vigor [117,118]. On the other hand, some researchers also reported that *Rht5* has negative effect over flowering time and delayed by 4.8–14% in a thermal environment [119,120]. In addition, there is still an attention required from molecular actors to identify the sequence of candidate genes through QTLs intervals, map-based cloning to enhance the lodging resistance in soybean.

### 3.2. Proper Sowing Time and Planting Density

Sowing time and planting density are both key factors in the maize-soybean intercropping system that affect the lodging resistance and yield of soybean crop [121]. Sowing time is a crucial factor that enhanced the competition within the species which ultimately reduced the crop yield [122]. Therefore, the selection of an optimal sowing time is vital perspective to enhance the lodging resistance and yield under the maize-soybean intercropping system [123]. It has been observed that delay sowing significantly decreased the risk of lodging by shortening the internodal length, plant height and center of gravity point and by increasing the cell wall thickness, diameter and grain filling period [124]. For instance, only two weeks late sowing could reduce up to 30% threat of lodging in wheat [125]. Under the maize-soybean intercropping the tall stature crops adversely affected the short stature crops at the both vegetative and reproductive stages as compared to relay intercropping in which the adverse effect could only be observed at vegetative stage [7]. However, in intercropping systems, the competition between the intercropped species could be decreased by fluctuating the planting time [126]. Most importantly, under relay intercropping systems, the crop sown first have more competition than the second crop [127]. On the other hand, under the relay intercropping system, selecting an optimal planting time could reduce the co-growth duration among the intercropped species and adverse effect of first sown crop could also be minimized on the second grown crop [128]. Furthermore, it is shown that fluctuating the planting time, 50 days of co-growth period of maize and soybean in relay intercropping system enhanced the crop growth rate 17–64% as compared to 70–90 days of co-growth period. In maize-soybean relay intercropping the soybean production was recorded maximum 2.11 t/ha at 50 days co-growth duration of maize and soybean [126].

High plant populations have been used extensively to increase crop production [77]. Under high planting densities, mutual shading of plants disturbs the light environment of crop which reduced the photosynthetic activities and carbohydrates accumulation in the stem that leads to lodging easily [25,129]. It has been witnessed that planting densities were negatively correlated with the lodging resistance of stem; however, high plant population had promoted the lodging and lower grain production [129,130,131,132]. In addition, optimum planting densities could expand the structure of the plant population and provide a better light environment, enhanced photosynthetic rate and ultimately increased the lodging resistance of stem [74,133]. Furthermore, it was found that in strip intercropping optimum planting densities that is, 17 and 20 plants/m^2^ had significantly enhanced the stem diameter by 4.3% and 6%, respectively as compared to 25 plants/m^2^ planting density. Most specifically, the plant height was decreased by 6.2% and 9.4% at 17 and 20 plants/m^2^, respectively than that of 25 plants/m^2^ planting density. In addition, decreasing the planting densities such as 17 and 20 plants/m^2^ could decrease the lodging percentage by 50.3% and 19.3%, respectively as compared to 25 plants/m^2^ planting density [134]. However, further field experiments to optimize the planting densities in the maize-soybean intercropping system Table 2.

### 3.3. Efficient Use of Fertilizers

Nitrogen (N), phosphorus (P) and potassium (K) are the important macro-elements, vital for crop growth and development [143]. N fertilizer application rate and time of application had also a significant effect on the lodging resistance of crops [144,145]. It has been described that high nitrogen rate had reduced the diameter of basal or lower internode that cause lodging [146]. On the other hand, the low application of nitrogen had enhanced the concentrations of water soluble carbohydrates as in the middle internodes 21% and in the basal internodes 42% than that of high nitrogen rate [147]. Several researchers suggested that a high rate of nitrogen application promoted vegetative growth and decreased the root anchorage in the soil and stem secondary cell wall composition (lignin content) which resulted in lodging of the crops [25,133,148]. A higher rate of nitrogen application had reduced the activities of lignin related enzymes (PAL, POD, CAD and 4CL) and lignin deposition in the cell wall which decreased the lodging resistance [24]. Berry et al. [149] also observed that a low rate of N had reduced the height of plant and also proliferated the stem diameter and cell wall width which turns in high stem strength. Another study had found that increasing the amount of N in wheat crop had progressively increased the cell wall constituents (lignin and cellulose content) and afterward it was decreased gradually [148]. Many recent studies showed that lodging resistance of crop could be improved at the price of yield sacrificing by minimizing the N application rate and rescheduling the application of N fertilizers [149,150]. Furthermore, the relation between N and K has a fundamental role in improving crop grain production and quality [146,150]. Increasing the level of K^+^ along with elevated NH4^+^ could decrease the stem cell wall thickness. For an instance, the application of a high level of N and P fertilizers in the absence of K decreased the 30%–35% grain production in rice due to lodging [151]. However, with the application of K, the stem mechanical strength could be increased [143]. An equal application rate of N and K had significantly promoted the root growth and enhanced the root anchorage which resulted in lodging resistance [152].

The optimum level of K^+^ nutrition to plants was positively correlated with lignin accumulation into the vascular bundles and cells of sclerenchyma of plant cell wall which ultimately enhanced the stem diameter and lodging resistance [153]. In the same way, it has been noticed that K^+^ considerably inhibited the adverse effects of a higher level of NH_4_^+^ which in turns amplified the 20–27% of stem mechanical strength in wheat [146]. However, K^+^ has a pivotal role in the process of photosynthesis and metabolic activities of carbohydrates production in plants [154,155]. However, appropriate fertilization of nitrogen could enhance the lignin content in the basal internode and improved the stem lodging resistance [24,79,156]. Conclusively, there is a still gap in nitrogen fertilization of soybean under maize-soybean intercropping; henceforth a more attention is required from agronomists and plant breeders for rescheduling the rate and application of nitrogen fertilizer which play a major role in lodging resistance and crop final production.

Previous experiments have revealed that the efficiency of intercrops to absorb nitrogen (N) is more than the sole-cropping, however, the total uptake of N by intercropped soybean and wheat is greater than the total of the sole crops [157,158]. According to an estimation, soybean can fix nitrogen about 39 to 182 kg N ha^−1^ [159]. It has been noticed that under high nitrogen conditions legumes are usually shaded by the maize which results in shade avoidance response and low grain production [160]. These adverse effects of cereal crops on legumes can be alleviated by fluctuating the sowing date [161,162]. A recent meta-analysis explained the land and fertilizer nitrogen use efficiency of intercropped maize and soybean [163]. They concluded that maize-soybean intercropping system has greater potential to attain high land equivalent ratio (LER) and fertilizer nitrogen equivalent ratio (FNER) by utilizing the optimum levels of nitrogen inputs. Whereas further studies are needed to pinpoint the optimum combinations of sowing configuration, planting dates and fertilizer rate and time to attain the high yields by reducing the lodging stress.

### 3.4. Development of Lodging Resistant Cultivars

Agronomists characterized the soybean cultivars into three sets depending on the genotypes response towards lodging resistance: highly, moderately and susceptible cultivars [164]. In some previously conducted experiments, different cultivars of soybean were grown under the maize-soybean intercropping system to distinguish the more suitable and lodging resistance cultivars. In a previous experiment, four recombinant inbred lines (B3, B15, B23 and B24) of Nandou-12 (that is shade tolerant and widely grown in maize-soybean intercropping system of China) and Nan 032-4 (that is shade susceptible cultivar) in were used [99], the lignin content in stem and lodging resistance index of B23 and B24 was significantly higher than that of B3 and B15 under both monocropping and intercropping systems. Furthermore, another experiment was conducted in which three cultivars were selected on the base of their response to lodging and shade stress [19]. A shade susceptible cultivar (Nan 032-4), a moderate cultivar to shade and lodging tolerance (Jiuyuehang) and shade tolerance and lodging resistance cultivar (Nandou-12) [19]. Their findings revealed that Nandou-12 had more accumulation of lignin in stem and high enzymatic activities of lignin related enzymes (PAL, 4CL, CAD and POD), hence more lodging resistance as compared to Nan 032-4 and Jiuyuehang cultivars under both monocropping and intercropping systems (Figure 3). Along with these research outcomes, there is a dire need to work out on the soybean cultivars which are well suited for intercropping systems with greater lodging resistance.

### 3.5. Role of Silicon and Titanium

Si plays a pivotal role in the growth and development of plants as a beneficial micronutrient. A significant impact of silicon has been observed on the plant height, internodal length and stem strength and lodging tolerance [165,166]. It was found that the application of Si could be dispersed from third to the fourth internode, which result in enhancement of lodging resistance [167]. A previous study revealed that Si had enhanced the thickness of the cell wall and the vascular bundles size in the rice stem [168]. With increase in the amount of Si application, the cellulose and hemicellulose content are increased which contribute in the cell wall formation of rice stem [169]. However, it was also identified that Si acts as ligands binding with hemicellulose in the rice cell walls [54,170]. A recent research showed that appropriate concentrations of Si could improve the enzymatic activity of lignin related enzymes and also prompted the gene expression of related enzymes [75]. It also revealed that Si content also increased the lignin content in the stem cell wall and promoted the lodging resistance of soybean under low light conditions. Moreover, Si also accelerated the photosynthetic activities, stomatal conductance and increased the chlorophyll content of tobacco under cadmium stress [171]. Wang et al. [172] observed that Si element could amplify the photosynthetic rate by altering the leaf structure and the content of chlorophyll in rice plants. In addition, Si element can change the leaf anatomy to capture more light and enhance the light interception in the plant’s leaves and to improve the vascular bundle sheath and sclerenchyma tissues, which help in lodging resistance [173,174]. Another research has concluded that Si could be used as fuel to ignite the process of lignification and silicification for the cell wall and collenchyma cells thickness that increases the development of keratinocyte and cellulose contents resulting in lodging tolerance [28].

On the other hand, the biological importance and role of titanium (Ti) in growth and development has been studied for decades but still, it is not recognized as an essential nutrient for the plants. However, recent research revealed that optimum concentration of Ti improves the leaf chloroplast structure, total biomass, chlorophyll fluorescence, root architecture, RuBisCO enzyme activity and total chlorophyll content of soybean plants under low light conditions [175]. In addition, how Ti affects the lignin content is not elaborated yet under low light conditions. Therefore, in future RNAseq transcriptional studies, proteomics and genomic profiling should be done to gain deeper insights into the effects and benefits of Si and Ti on soybean stem strength, lodging resistance under the maize-soybean intercropping system.

### 3.6. Chemical Approaches

Foliar application of plant growth regulators at the suitable growth stage of a crop can enhance the stem mechanical strength, reduce the plant height and inhibit lodging [176,177]. Plant growth hormones that stop the biosynthesis of GA are being widely used in high input cropping systems to decrease straw content and also increase the lodging resistance [176]. Many plant growth regulators were extensively utilized to minimize lodging index through shortening the plant height and to obtain a stable grain production [177,178]. The most common growth regulators that inhibit the GA production have been used to decrease the growth of stem are the onium type elements, which have Chlormequate chloride (2-chloroethyl*-N, N, N-*trimethyl-ammonium chloride, CCC) and Mepiquat-Cl. Some other plant growth regulators which comprise N heterocycles for example, triazoles and imidazoles could also be used to minimize the lodging risk [179]. The application of paclobutrazol had considerably increased the lignin content in the stem cell wall and its function is closely related to enzymes present in the basal second internode [180]. It had also enhanced the stem diameter, internode filling degree and cell wall thickness with increasing the lodging resistance [71]. The Paclobutrazol chemical (PP333) could prompt the enzymatic activities of lignin related enzymes that is, tyrosine ammonia-lyase (TAL), phenylalanine ammonia-lyase (PAL) and cinnamyl alcohol dehydrogenase (CAD) and ultimately proliferated the lignin content and lodging resistance [181]. A plant growth regulator (Trinexapac-ethyl) has reduced the height of the plant in wheat [182,183]. In addition, some other chemicals are needed to be introduced which are capable of enhancing the lodging resistance in intercropped soybean to get better results.

## 4. Future Prospects

In the maize-soybean intercropping system, shade has a drastic effect on the normal growth and development of soybean at both vegetative and reproductive stages. Shading of maize disturbs the microclimate of soybean which results in shade avoidance response of soybean (stem elongation) and finally causes lodging. The influence of lodging concerning the reduction of grain yield depends on the types of cultivars and planting geometries. This review mainly focused on the multiple approaches and genetic techniques which would be helpful to control the lodging under intercropping systems (Figure 3). The damages induced by lodging in maize-soybean intercropping could be actively reduced with more advanced crop breeding techniques. We further explored that, lignin and cellulose are the main constituents of plant cell wall which play a vital role in plant vigor against biotic and abiotic factors such as lodging. However, the molecular mechanism of lignin and cellulose formation and their relation with hormones (indole-acetic acid, IAA) that how they affect each other has not been explored thoroughly yet. Therefore, it is necessary to explore the biochemical association of hormones and lignin biosynthesis pathways in the maize-soybean intercropping system.

In addition, researchers have to develop new techniques and tools to modify the lignin content in soybean stem without altering its normal functions. In consequences of natural calamities like high-speed winds and rainfall could damage the crops catastrophically through lodging. Therefore, to escape from these devastating effects of lodging following approaches could be adopted: (i) breeding of soybean cultivars with stronger and harder roots without disturbing the existing root numbers per plant, (ii) suitable agronomic management’s that is, use of lodging resistant cultivars, rescheduling the planting time and density and use of fertilizers. Furthermore, plant growth regulators can also manipulate the height of the plant which helps in lowering the risk of lodging. Optimum levels of N, P, K and Si fertilizers could play a significant role in the maize-soybean intercropping system. Additionally, further studies are required to alter the plant canopy area which is a vital part of modern agronomy techniques and it is usually obtained through the application of balanced nitrogen fertilizers.

## Figures and Tables

**Figure 1 plants-09-01592-f001:**
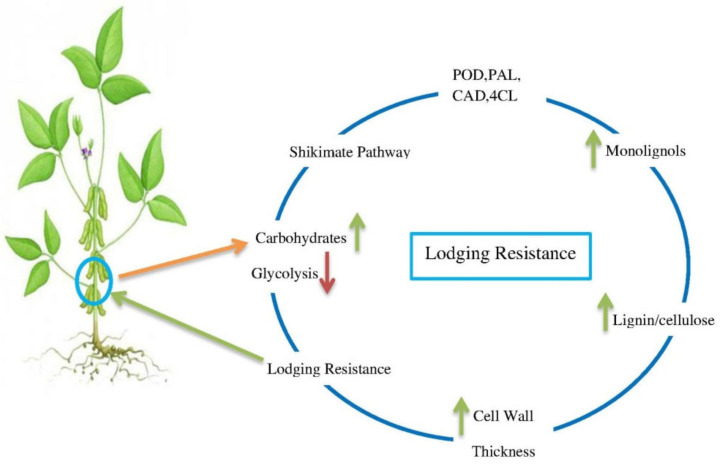
Role of carbohydrates and lignin enzymes in lodging resistance, green arrows represented the increasing trend while red arrows represent the decreasing trend.

**Figure 2 plants-09-01592-f002:**
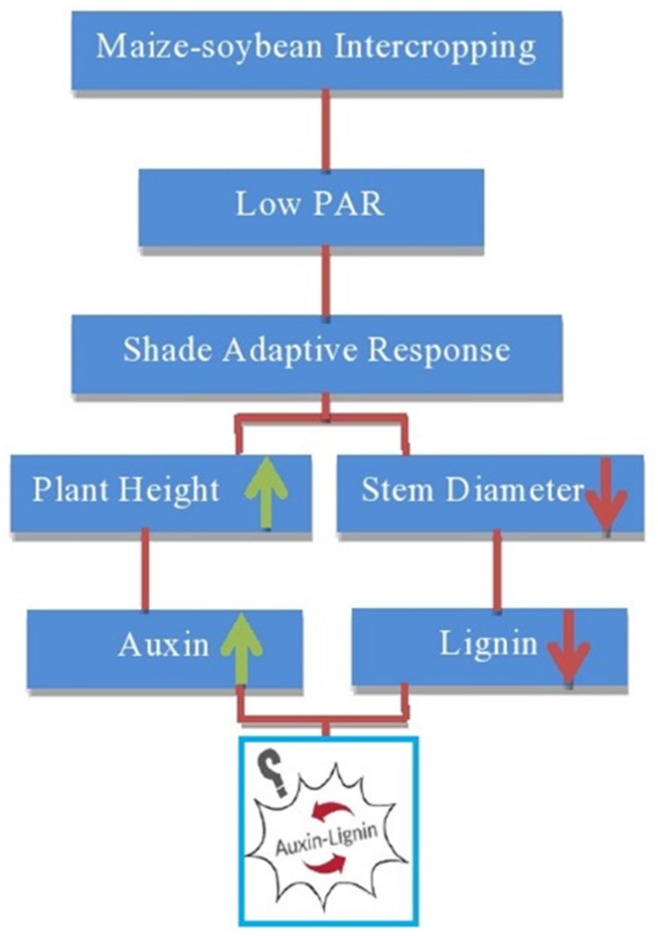
Shade adaptive response of soybean to low light intensity and relationship between phytohormone (auxin) and lignin.

**Figure 3 plants-09-01592-f003:**
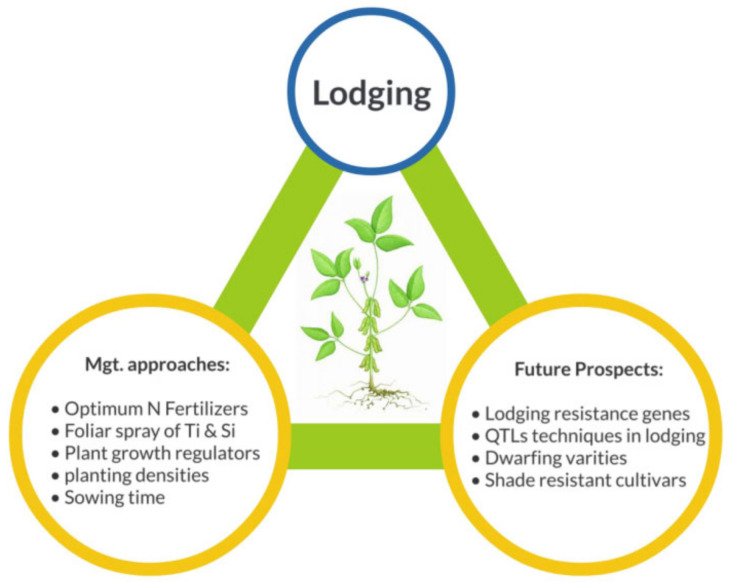
An overview of the most suitable approaches including molecular breeding techniques, agronomical managements and chemical controls to mitigate the lodging stress in maize-soybean intercropping.

**Table 1 plants-09-01592-t001:** Soybean constituents correlated with lodging resistance under maize-soybean intercropping system.

Trait (s)	Cropping System	Behavior	Reference (s)
Morphological Traits	**Maize-soybean Intercropping**		
Plant Height		Positively Correlated with lodging	[48]
Internodal-length		Negatively correlated with lodging resistance	[47,73]
Stem Diameter		Strongly correlated with resistance to lodging	[40,74]
Anatomical Features			
Number of vascular bundles		Positively correlated with lodging resistance	[75]
Width of vascular bundle sheath		Strongly associated with resistance to lodging	[51,76]
Cell Wall Thickness		Strongly correlated with lodging resistance	[59]
Physical Aspects			
Wind		Significantly associated with lodging	[40]
Rainfall		Strongly correlated with lodging	[74]
Shade		Negatively associated with lodging resistance	[59,77,78]
Biochemical Features			
Lignin and Cellulose Content		Positively associated with cell wall thickness and lodging resistance	[19,79]

**Table 2 plants-09-01592-t002:** Comparative analysis of maize-soybean intercropping system (MSIS) in different countries.

	Planting Density in Monocropping (×10^3^ Plants ha^−1^)	Planting Density in Intercropping (×10^3^ Plants ha^−1^)	
Country	Maize Soybean	Maize Soybean	Reference (s)
China (MSIS)	59 117	59 117	[124,134,135]
Egypt	71 143	24 95	[136]
Ethiopia	44 500	44 375	[137]
Ghana	56 111	56 222	[138]
India	83 333	42 250	[139]
Nepal	40 200	40 200	[140]
Nigeria	33 200	33 200	[141]
Iran	36 400	36 400	[142]

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
