# Peer review of "Agro-Techniques for Lodging Stress Management in Maize-Soybean Intercropping System—A Review"

_plants, 2020, doi:10.3390/plants9111592_

Round 1
Reviewer 1 Report
In the cultivation of cereals, lodging is a serious problem.It causes quantitative and qualitative losses in the grain yield, hinders
the mechanical collection of collection points to increase costs with it.
The amount of losses depends on the degree of lodging and the timing of its
occurrence.
The authors presented a publication describing the problem of lodging in
maize very well.
This publicationis well written and in my opinion is acepted in present
form.
Author Response
Response Letter
Dear editor,
Manuscript ID; plants-971200 entitled “Agro-techniques for Lodging Stress Management in Maize-soybean Intercropping System-A review ” is re-submitting after minor revision to the “Plants” journal for consideration of publication as a regular paper.
We do appreciate the comments from you and reviewers and considered carefully. Please do find a revised version of our manuscript. Later in this text we responded to each and every comment of the reviewers. We hope that the revisions in the manuscript and our accompanying responses will be sufficient to make our manuscript suitable for publication in Plants. We believe that our study will be interesting for the readers of Plants and as well as for a broad community of plant scientists.
Yours sincerely,
Liu Weiguo
Author’s response to reviewers
Reviewer #1
Dear reviewer,
Thank you so much for your time and appreciation. The authors are much grateful and pleased for this kind of support of critically reviewing our manuscript.

Reviewer 2 Report
This manuscript reviewed different agro-techniques for lodging stress management in maize soybean intercropping systems, which was very interesting to me to read. The authors presented different agronomical, biochemical and genetic approaches very nicely. However, as a reviewer, I have some suggestions which authors may consider to improve the manuscript for publication.
Line 152-153: this line have been mentioned several times in previous sections. See line 78-79, line 127-128
Line 154-155: this finding also repeated several times in different sections. See line 81-82, line 129-130. Try to avoid repetition.
Line 149-150 already mentioned in line 124-125.
Line 156-157 already mentioned in line 130-131. Please omit repetition.
Line 195: “stem physical strength of stem”. You wrote ‘stem’ twice. Delete first one.
Line 281: is it “lodging resistance genes to lodging susceptible genotypes”? or lodging susceptible genes to lodging resistance genotypes ?
Line 308-309: this finding may be true for wheat monoculture but such late sowing also have potential risk of reducing wheat grain yield due to short growing season.
Line 314: competition for what? In relay intercropping no competition exist until second crop grown !!
Line 339: sentence incomplete. …… in different ** ?
Line 344: Delete ‘the’ after ‘diameter of basal or lower’. Write “…….. diameter of basal or lower internode…..”
Line 361: ‘30-35% grain production in rice….’ Is it increase or decrease?
In line 359-360, you wrote increasing level of K decrease the stem cell wall thickness, again in line 362 you mention that application of K increase stem mechanical strength. Is not it contradictory?
Figure 4 not directly related with the discussion. This figure shows Lignin biosynthesis pathway of E93 cultivar which was not mentioned in the text. The authors should remove this figure.
Author Response
Response Letter
Dear editor,
Manuscript ID; plants-971200 entitled “Agro-techniques for Lodging Stress Management in Maize-soybean Intercropping System-A review ” is re-submitting after minor revision to the “Plants Journal” for consideration of publication as a regular paper.
We do appreciate the comments from you and reviewers and considered carefully. Please do find a revised version of our manuscript. Later in this text we responded to each and every comment of the reviewers. We hope that the revisions in the manuscript and our accompanying responses will be sufficient to make our manuscript suitable for publication in MDPI PJ. We believe that our study will be interesting for the readers of MDPI PJ and as well as for a broad community of plant scientists.
Yours sincerely,
Liu Weiguo
Author’s response to reviewers
Reviewer #2:
- Line 152-153: this line have been mentioned several times in previous sections. See line 78-79, line 127-128.
Response:
First of all we would like to thank you for your appreciations and efforts toward our manuscript. We have omitted the repeated lines from the manuscript according to your suggestions. To see the modification please see the line numbers 78-79 and 127-128.
- Line 154-155: this finding also repeated several times in different sections. See line 81-82, line 129-130. Try to avoid repetition
Response:
We have omitted the repeated findings from the following sections, to see please check the line numbers 128-130 and 153-155.
- Line 149-150 already mentioned in line 124-125.
Response:
We have omitted the following repeated line. To see modification please see the line numbers 149-151.
Line 156-157 already mentioned in line 130-131. Please omit repetition.
Response:
We have omitted the repetition of lines. Please check the line numbers 130-132.
- Line 195: “stem physical strength of stem”. You wrote ‘stem’ twice. Delete first one. Response:
We have modified this sentence. To see please see the line number 195.
- Line 281: is it “lodging resistance genes to lodging susceptible genotypes”? or lodging susceptible genes to lodging resistance genotypes ?
Response:
We have modified this sentence according to your suggestion. To check please see the line number 281.
- Line 308-309: this finding may be true for wheat monoculture but such late sowing also have potential risk of reducing wheat grain yield due to short growing season.
Response:
Yes, there is a risk of reduction in wheat grain production, but this practice could be tried in maize-soybean relay intercropping, likewise it is described in line numbers 317-321 (Furthermore, it is shown that fluctuating the planting time, 50 days of co-growth period of maize soybean in relay intercropping system enhanced the crop growth rate 17%-64% as compared to 70-90 days of co-growth period. In maize-soybean relay intercropping the soybean production was recorded maximum 2.11 t/ha at 50 days co-growth duration of maize and soybean) [1]. If we sow the soybean late then the co-growth days of maize and soybean will be decreased which minimized the shading duration of maize on soybean and ultimately soybean growth would be better.
- Line 314: competition for what? In relay intercropping no competition exist until second crop grown !!
Response:
Yes, you’re right in relay intercropping no competition exists until second crop grown. But we’re talking about the competition after second crop grown, likewise, in maize-soybean relay intercropping due to firstly grown, tall stature and more canopy area of maize plant which prevent the light interception to small stature of soybean plants. So, there is competition between light and nutrient uptake due to already root growth and expansion of maize plants which cause the poor growth of soybean.
- Line 339: sentence incomplete. …… in different ** ?
Response:
We have completed the following sentence. To check please see the line number 339.
- Line 344: Delete ‘the’ after ‘diameter of basal or lower’. Write “…….. diameter of basal or lower internode…..”
Response:
We have modified the sentence. To see modification please see line number 344.
- Line 361: ‘30-35% grain production in rice….’ Is it increase or decrease?
Response:
It decreases the 30-35% grain production in rice. We have revised this sentence and also add “decrease” in the sentence. To see modification please see the line number 361.
- In line 359-360, you wrote increasing level of K decrease the stem cell wall thickness, again in line 362 you mention that application of K increase stem mechanical strength. Is not it contradictory?
Response:
Thank you very much for your careful reading and comments to improve our manuscript. In this finding, previous studies revealed about the imbalance fertilization of N and K like described with an example in line numbers 361-362. Likewise if both N and K levels are high this is caused higher plant height and reduces stem thickness. According to previous studies balanced application of K could enhance the stem breaking strength. Here’s the reference is cited [2].
- Figure 4 not directly related with the discussion. This figure shows Lignin biosynthesis pathway of E93 cultivar which was not mentioned in the text. The authors should remove this figure.
Response:
Thank you for your valuable suggestion. We have removed this figure 4 from the manuscript. To see modification please see the line numbers 407-409.
Reference
Ahmed, S.; Raza, M.A.; Yuan, X.; Du, Y.; Iqbal, N.; Chachar, Q.; Soomro, A.A.; Ibrahim, F.; Hussain, S.; Wang, X., et al. Optimized planting time and co‐growth duration reduce the yield difference between intercropped and sole soybean by enhancing soybean resilience toward size‐asymmetric competition. Food and Energy Security 2020, 9, doi:10.1002/fes3.226.- Shah, L.; Yahya, M.; Shah, S.M.A.; Nadeem, M.; Ali, A.; Ali, A.; Wang, J.; Riaz, M.W.; Rehman, S.; Wu, W., et al. Improving Lodging Resistance: Using Wheat and Rice as Classical Examples. Int J Mol Sci 2019, 20, doi:10.3390/ijms20174211.
